# Effect of Olive Fruit Volatiles on Landing, Egg Production, and Longevity of *Bactrocera oleae* Females under Different Temperatures

**DOI:** 10.3390/insects15090728

**Published:** 2024-09-21

**Authors:** Anastasia Kokkari, Nikos A. Kouloussis, George Floros, Dimitrios S. Koveos

**Affiliations:** Laboratory of Applied Zoology and Parasitology, School of Agriculture, Aristotle University of Thessaloniki, 54124 Thessaloniki, Greece; akokkari@agro.auth.gr (A.K.); nikoul@agro.auth.gr (N.A.K.); florosgd@agro.auth.gr (G.F.)

**Keywords:** olive fruit, volatile organic compounds, olive fly, *Bactrocera oleae*, egg production, longevity, landings on fruit

## Abstract

**Simple Summary:**

The olive fruit fly, *Bactrocera oleae* Gmelin (Diptera: Tephritidae), is one of the most harmful olive pests in Mediterranean countries and worldwide, and its control is mainly based on chemical insecticides. Volatile organic compounds (VOCs) emitted from the olive fruit affect the physiology and behavior of the olive fly and could be an effective tool for its control. In this study, we determined the effect of certain olive fruit VOCs on attraction to olives, egg production, and longevity of *B. oleae* females, under a series of temperatures. The results show that the exposure of female flies to certain VOCs either increased or decreased the number of landings on the olives, egg production, and longevity. These findings are poised to advance our understanding of the role of olive fruit volatiles on olive fly behavior and may contribute to the improvement of bait or repellent products for its control.

**Abstract:**

Females of the olive fruit fly *Bactrocera oleae* use various contact and volatile plant stimuli to find olive fruits and lay their eggs on them. We detected certain volatile organic compounds (VOCs) emitted from the olive fruit and studied their effect on female landings on olive fruits, egg production, and longevity under a series of different temperatures from 15 °C to 35 °C. When female flies were maintained at temperatures from 17 °C to 30 °C and exposed to different fruit VOCs either increased or decreased, depending on the substance tested, their landings on olives, egg production, and longevity. Temperature significantly affected the females’ responses to fruit VOCs. The highest responses of the flies to fruit VOCs were observed at 30 °C, except for longevity. By contrast, at 15 °C or 35 °C, the flies did not show any response to VOCs. Our results may contribute to a better understanding of the olive fly positive or negative responses to fruit VOCs and the improvement of its control.

## 1. Introduction

The olive fruit fly *Bactrocera oleae* (Rossi) (Diptera: Tephritidae) is the major pest of olive orchards worldwide, causing severe economic losses to olive growers [1,2]. It is a monophagous pest and its larva feed, exclusively, on the mesocarp of olive fruits causing fruit damage, premature fruit drops, and oil deterioration [3]. Preimaginal stages of the olive fruit fly develop in the olive fruit, during summer, spring, and autumn or winter if temperature enables it [4,5,6,7].

The adult olive fly’s behavior is strongly dependent on its host fruit and related to the fruit’s physical and chemical stimuli. The olive fruit favors the ovarian maturation of the females due to contact or volatile stimuli. Contact stimuli favoring ovarian maturation could be waxes of the olive epicarp and bacteria present on the fruit’s surface [8,9,10,11,12]. In addition, olive fruit, due to its stimuli (contact and volatile), favors mating success and egg production of the olive fruit fly [13]. Yet, exposure of the olive fruit fly females to *α*-pinene increased their fecundity [14]. In previous work, we determined that many volatile compounds emitted from cv. Megaritiki, one of the Greek olive cultivars, favored egg production of the fly [15]. Olive varieties vary in their susceptibility to *B. oleae* infestation [16], which is primarily due to the females’ oviposition preferences and the fruit’s suitability for larval development. Adult females carefully select suitable ovipositional sites relying on olfactory cues emitted by the olive fruits [17]. Gonçalves et al. [18] have found that among the main olive varieties cultivated in Portugal, Cobrançosa is less susceptible to olive fly injury.

Plant-emitted volatile organic compounds (VOCs) act as chemical signals (kairomones) interacting with herbivore insects [19,20,21,22]. It is known that more than 1700 volatile products, such as alcohols, aldehydes, and ketones, are released from plants [23]. Furthermore, volatiles emitted from host plants may increase the longevity of herbivorous insects and can act as a food additive. Various stimuli such as food odor, pheromones, and temperature affect the longevity and reproduction of insects [24,25,26]. Food-derived odorants altered the lifespan of *Drosophila melanogaster*, which was reduced under dietary-restricted conditions and extended when the flies were fully supplied with food [27].

Herbivorous insects’ physiology and behavior are influenced by changes in the emitted volatiles of their host plants [28]. These variations in plant VOC emissions may be caused by temperature changes, which, also, can directly affect herbivorous insects [29]. Hence, the effects of high temperatures on plants’ VOCs are crucial for the interactions between plant and their environment. Rising temperature significantly affects the composition and concentration of secondary plant metabolites, including the VOCs, and subsequently affects the behavior and attacks by phytophagous insects [30,31,32]. Therefore, increasing ambient temperature can affect the susceptibility of plants to attacks by herbivorous insects [33]. As Zvereva and Kozlov [34] showed the temperature affects the quantity as well as the nutritional quality of foliar and plant tissues for herbivorous insects. More specifically, increased temperatures resulted in reduced sugar and starch levels, whereas the nitrogen concentration of the tissue remained untouched. In addition, many volatile substances released by plants contribute to their resistance to heat stress [35]. Linalool and ocimenes are monoterpenes emitted from plants under stressed conditions, such as high temperatures [36,37]. Similarly, higher temperatures affect the production, in higher amounts, of phenolic compounds, including tannins, flavonoids, and terpenoids, in the plant *Robinia pseudoacacia* [38]. Interestingly, changes in the emitted amounts of the VOCs and temperature may alter the food-plant preference of herbivorous insects. For example, when potato and nightshade (*Solanum dulcamara*) plants were offered as food to Colorado potato beetles (*Leptinotarsa decemlineata*), they normally showed a preference for potatoes. This was reversed at 25 °C or higher temperatures. Temperature may also modify the insects’ behavior through changes in their chemoreceptors or central nervous system [39].

Temperature is one of the most important factors that affect the insect’s survival, reproduction, abundance, distribution, and fitness in their environment [40,41,42]. In certain regions of Greece and Italy, adult female olive fruit flies are reproductively immature, during spring and early summer. This immaturity of the females may be caused by the increasing temperature and changing photoperiod that occurs during this period of the year [7,43,44,45,46,47,48,49]. Temperature and photoperiod affect ovarian maturation under laboratory conditions [10,11]. A reproductive immaturity was induced in females of the olive fly, when they developed in the preimaginal stages at 19 ± 1 °C and 12:12 L:D and subsequently maintained as adults at 26 ± 1 °C and 16:8 L:D. However, this ovarian immaturity was averted when the flies had access to olive fruit [11]. Similarly, Fletcher et al. [8] showed that in Corfu, Greece, during late summer, the presence of olive fruits favored the olive flies’ ovarian maturation whereas low humidity and high temperatures inhibited it. Besides that, high summer temperatures are a boundary for the development of the olive fly, affecting its dispersal ability, survival, and oviposition [50,51].

The effect of high temperatures during summer on the ovarian maturation of the female olive flies has been investigated by many researchers [7,8,52,53], as well as the effect of olive fruit on *B. oleae* behavior [10,11,12,13,15,17,54,55,56,57]. However, the effect of temperature on the attractiveness of olive fruit VOCs and their impact on female egg production remains unidentified. In this research, we investigated, under laboratory conditions, the impact of certain olive fruit VOCs (α-pinene and limonene) and 3 VOCs mixtures (*n*-octane–*α*-pinene–limonene–ethylhexanol–nonanal–*n*-dodecane–decanal–*n*-tetradecane (2:12:2:3:2:1:1:2); mixture 2: *n*-octane–ethyl hexanol, 4:1; and mixture 3: limonene–nonanal, 1:1) on the number of female landings on the olive fruit and egg production of females, under a series of temperatures. Moreover, we tested whether limonene and mixture 1 of VOCs affect female longevity and egg production. The rationale behind these experiments was to determine whether VOCs emitted from olive fruits affect the fly’s behavior and physiology under different temperatures and thus being useful for the improvement of the control and mass rearing of the fly.

## 2. Materials and Methods

### 2.1. Stock Colony and Experimental Flies

The stock colony was established with adult flies that emerged from field-infested olive fruits collected in the area of northern Greece (Chalkidhiki). The stock colony flies were maintained in wooden cages (30 × 30 × 30 cm) with three sides of wire screen, in a climatic room with a temperature of 25 ± 1 °C, relative humidity (RH) of 65 ± 1%, and photoperiod of 16:8 L:D. A liquid diet consisting of water, yeast hydrolysate, and commercial sucrose (volume–weight–weight ratio of 5:4:1) was offered to the flies. The stock colony flies had access to water via a wet cotton wick placed in a polyprene vial. Stock colony females emerged from the infested field collected olives were allowed to lay a single egg in each olive fruit, and subsequently, the emerging flies were used in the experiments. The olive fruits used in the experiments were collected in early August and maintained at 5 ± 1 °C for a period of 2–3 months before their use in the experiments.

### 2.2. Effect of Olive Fruit Volatiles on Females’ Landings on Olives under Different Temperatures

We studied the effect of different temperatures on the responses of female flies to certain VOCs. For the experiments, we maintained the adult flies in cages exposing them to certain synthetic VOCs and VOC mixtures at different temperatures and recorded their landings on olives. The tested VOCs were the following: *α*-pinene; limonene; mixture 1, consisted of *n*-octane–*α*-pinene–limonene–ethyl hexanol–nonanal–*n*-dodecane–decanal–*n*-tetradecane (12:12:2:3:2:1:1:2); mixture 2, consisted of *n*-octane–ethyl hexanol (4:1); mixture 3, consisted of limonene–nonanal (1:1). In all the cases, the tested olive fruit VOCs were diluted in acetone in a ratio of 1:1. We selected to use the above-mentioned VOCs and mixtures because in a previous work [15], we found that they were emitted in the highest relative percentages by mature olive fruits and caused the most pronounced impact (positive or negative) on egg production and mating. The chemical ratio of the compounds in the tested mixtures was estimated based on our previous results of GC-MS analysis of olive fruit volatiles [15].

The pure chemicals dodecane, 2-ethylhexan-1-ol, *n*-octane, D-limonene, decanal, and 1-nonanal were obtained from Fluorochem Ltd.; Tetradecane from Sigma-Aldrich Gmb; and *α*-pinene from Alfa Aesar (Karlsruhe, Germany). The purity of the compounds was greater than 98%.

For the experiments, 10-day-old mated females were maintained, in groups of 10, in plexiglass cages (20 × 20 × 20 cm), with their sides bearing a hole covered with mesh cloth for ventilation and exposed to the tested VOCs at 15, 17, 20, 25, 30, 33, and 35 °C in climatic incubators with a photoperiod of 16:8 L:D and relative humidity (RH) 65 ± 1%. The tested VOCs or their mixtures were applied in two quantities (5 μL or 20 μL) on a filter paper disk (diameter: 3 cm) (Whatman filter paper, Sigma-Aldrich Gmb, Burlington, Massachusetts, United States) that was hung on the top of each cage with a thin spinner. An olive fruit was hung below the VOC-soaked filter paper in the same spinner. We considered that the more attractive the VOCs, the more times the flies landed on the olive fruits. As a control, we used filter papers treated with only acetone in the same quantities (5 μL or 20 μL) as the VOCs.

We observed the flies in each cage every hour during the photophase and scored the number of flies that landed on the olives. There were four replicates (cages with 10 females) for each tested VOC or mixture of VOCs.

### 2.3. Effects of Olive Fruit Volatiles, under Different Temperatures, on Egg Production and Longevity

In this group of experiments, 10-day-old, mated females were transferred and maintained throughout their adult life in plexiglass cages (20 × 20 × 20 cm) in groups of 10. The flies in the cages were continuously exposed to either mixture 1 or limonene following the same experimental procedure as described above, at a series of different temperatures of 15, 17, 20, 25, 30, 33, and 35 °C, a photoperiod of 16:8 L:D, and relative humidity (RH) of 65 ± 1%. Egg production was scored every two days, by counting the number of oviposition holes in the olives, with the use of a stereoscope (Zeiss Stemi 305^®^, Oberkochen, Germany). Every day we scored mortality and renewed olive fruits and VOCs. There were four replicates (cages with 10 females) for either mixture 1 or limonene at each tested temperature.

We selected to use mixture 1 of VOCs and limonene because, in earlier experiments described in the above-mentioned paragraph, they were found to have the most pronounced positive or negative effect, respectively, on the number of flies’ landings on olives. In addition, earlier work of our group has shown that mating percentages significantly increased or decreased after exposure of the flies to mixture 1 or limonene, respectively [15].

### 2.4. Statistical Analysis

A two-way ANOVA was performed to detect whether the tested VOCs, temperature, and their interaction had a significant effect on the number of female landings on olives, egg production, and longevity.

To perform multiple comparisons of means for the evaluation of the attractive effects of olive fruit’s VOCs on the number of flies’ landings on the olives during a day at various temperatures, we further recorded our data set as a one-way ANOVA by using 49 levels of one pseudo-factor condition for the first series of our experiments and 28 levels for the second one (i.e., combination treatment; VOCs and temperature). Subsequent significant differences (*p* ≤ 0.05) of means were further separated with the Tukey HSD test. Before data analysis, Kolmogorov–Smirnov’s and Levene’s tests were used to confirm normality and homogeneity of variances. When heterogeneity of variances was significant, appropriate data transformation (log transformation) was performed prior to analysis. In cases, in which the latter transformation did not satisfy the parametric analysis criteria, Kruskal–Wallis non-parametric variance analysis, followed by the Mann–Whitney *U* test for all possible pairwise comparisons, was used (IBM SPSS 25.0) (IBM Corp., Armonk, NY, USA). The significance level was set at α = 0.05 (*p* ≤ 0.05).

## 3. Results

### 3.1. Effect of Olive Fruit Volatiles on the Number of Females’ Landings on Olives

Two-way ANOVA demonstrated that VOCs (F_5,126_ = 9.12, *p* < 0.001) and temperature (F_6,126_ = 462.25, *p* < 0.001), as well as their interaction (F_30,126_ = 462.25, *p* < 0.001), had a significant effect on the number of females’ landings on olives.

Table 1 shows that after their exposure to mixture 1 of VOCs, females significantly increased their landings on the olive fruits at the range of temperature from 17 °C to 30 °C (F_5,18_ = 99.12, *p* < 0.001). A similar positive effect on the number of female landings on the fruit was observed after exposure to α-pinene. By contrast, after females’ exposure to limonene or mixture 3 of VOCs, there was a significant decrease in the number of their landings on the fruits. The highest positive effect was observed after exposure to volatiles of mixture 1, i.e., 76.50 ± 1.04 landings compared to 25.75 ± 1.31 landings in the control. The highest negative effect was observed after exposure of females to limonene at 25 °C, i.e., 2.25 ± 0.25 landings compared to 24.00 ± 1.31 landings in the control.

At temperatures of 15 °C and 35 °C, the flies were not able to fly and, therefore, none of them visited the olives irrespective of the presence of VOCs. The flies’ activity and the number of landings on the olives, increased progressively with the temperature increase from 17 °C to 30 °C and decreased at 33 °C.

Hence, it is concluded that certain VOCs such as mixture 1 and α-pinene significantly increased the number of female landings on the olives. By contrast, mixture 3 and limonene significantly decreased the number of landings on the fruits at 17, 20, 25, and 30 °C, compared to the control ones. At temperatures of 15, 33, and 35 °C, the number of female landings on the fruits was very low to zero, in all the treatments and the control.

### 3.2. Effect of Olive Fruit Volatiles on the Number of Eggs Laid in Olives

In this group of experiments, we exposed the female flies during their adult life to either mixture 1 of VOCs or to limonene at a series of different temperatures and subsequently scored egg production. Two-way ANOVA demonstrated that VOCs and temperature as well as their interaction significantly affected the number of eggs laid in the olives (VOCs: F_3,96_ = 3296.36, *p* < 0.001; temperature; F_7,96_ = 1292.23, *p* < 0.001; interaction: F_21,96_ = 1253.35, *p* < 0.001). Exposure of the females to mixture 1 of VOCs significantly increased the number of eggs laid in the olives at temperatures from 17 °C to 33 °C (Figure 1, Table 2). By contrast, exposure of the females to limonene significantly reduced the number of eggs in all the tested temperatures (Table 2). The highest negative effect on egg production was observed after females’ exposure to limonene at 30 °C (102.75 ± 8.67 eggs compared to 994.75 ± 13.00 in the control) (Table 2). At 15 and 35 °C, females did not lay eggs (Table 2).

As shown in Figure 1, the oviposition period was prolonged after females exposure to mixture 1 of VOCs at 17, 20, and 33 °C (Figure 1). More specifically, females exposed to mixture 1 continued to lay eggs until the 48th, 56th, and 48th day of their life at 17, 20, and 33 °C, respectively, whereas non-exposed females (control) laid eggs until the 44th day of their life.

Our results show that a specific mixture of VOCs caused a significant increase in egg production, whereas limonene reduced the egg production of the flies. The intensity of these effects depends on the prevailing temperature.

### 3.3. Effect of Olive Fruit Volatiles (VOCs) on Female Adult Longevity

Two-way ANOVA showed that temperature (F_7,96_ = 1292.23, *p* < 0.001), VOCs (F_3,96_ = 1298.39, *p* < 0.001), and their interaction (F_21,96_ = 33.14, *p* < 0.001) significantly affected the longevity of adult females. As shown in Table 3, the longevity of females exposed to the mixture of VOCs was significantly reduced at temperatures of 25, 30, 33, and 35 °C, but it was not affected at the lower temperatures of 15 and 17 °C. After exposure to limonene, longevity was significantly reduced at all the tested temperatures. The highest negative effect on longevity was observed after exposure of the flies to limonene at 35 °C, i.e., 13.00 ± 0.58 days compared to 52.00 ± 0.82 days in the control.

Females lived longest at 15 °C and shortest at 35 °C irrespective of their exposure to VOCs (Table 3).

## 4. Discussion and Conclusions

Our present results show that a mixture of VOCs consisted of *n*-octane–*α*-pinene–limonene–ethyl hexanol–nonanal–*n*-dodecane–decanal–*n*-tetradecane 12:12:2:3:2:1:1:2 and α-pinene favored female landings on the olive fruit and egg production at different temperatures. By contrast, exposure of females to limonene significantly reduced the number of female landings on the olives and egg production at all the tested temperatures. The favorable effect or the non-favorable effect of certain VOCs was more pronounced at 30 °C, which may be due to the way flies perceive VOCs through their sensory organs. However, we cannot exclude the possibility that the effect of VOCs is likely to be enhanced by the presence of the olive fruit. The tested synthetic volatiles may act synergistically with the volatiles released from the olive fruit, resulting in a significant increase or decrease in the number of female flies landing on the fruits [58]. In addition, the flies could feed on the olive fruit surface after landing and have access to bacteria which act as protein sources, favoring egg production [59]. Similarly to our work, other VOCs such as α-copaene [56] and α-pinene [14,52] acted as ovipositor promoters that induced oviposition of *B. oleae* females. Jayanthi et al. [60] found that the volatile compound *γ*-octalactone favors the oviposition response in a related species, namely, *Bactrocera dorsalis* (Diptera: Tephritidae).

Moreover, previous studies have shown that olive fruit had a stimulatory effect on ovarian maturation via contact and volatile stimuli [9,10,11,13]. Females of *B. oleae* use olive fruit volatiles as stimuli to lay their eggs in the olive fruit. Exposure of the flies to these chemicals favored mating and egg production of the flies [15]. Olive fruit VOCs may be related to fruit susceptibility to olive fly infestation. However, other factors may play a vital role such as the maturity index, weight, and volume [18,46], fruit color [46,61,62], surface waxes [63], and fatty acid composition [18]. In a previous work, we determined >80 volatile compounds distributed in different chemical classes, emitted from olive fruits [15]. The composition of the olive fruit VOCs is affected by the ripening degree of the olive fruit [15,54,56] and the variety of the olive tree [18]. It has been demonstrated that the release of olfactory cues from mature host fruits can influence the oviposition behavior of Tephritidae flies [64,65]. Females of the olive fly prefer to lay eggs in August’s than June’s green olive fruits [66], which may be related to the synthesis and emission of certain VOCs during fruit ripening, which could stimulate egg production and oviposition [67].

In the current study, we demonstrated that a mixture of olive fruit VOCs, as well as limonene, affected the longevity of *B. oleae* females at certain temperatures. Our results show that a mixture consisting of *n*-octane–*α*-pinene–limonene–ethyl hexanol–nonanal–*n*-dodecane–decanal–*n*-tetradecane (12:12:2:3:2:1:1:2) did not affect longevity, at 15, 17, and 20 °C, whereas it caused a significant reduction in longevity at 25, 30, 33, and 35 °C. In addition, exposure of *B. oleae* females to limonene significantly reduced longevity at all the tested temperatures. Similarly to our study, Gerofotis et al. [14] demonstrated that olive fruit volatiles affect the olive fly’s lifespan. In particular, the exposure of *B. oleae* adult flies to *α*-pinene, released from olive fruits, increased the lifespan of male olive flies that reared under conditions of food deprivation [14]. Poon et al. [27] showed that carbon dioxide seems to extend the longevity of *Drosophila melanogaster* (Diptera: Tephritidae). The production of insects’ hormones that regulate physiological changes, such as females’ various reproductive functions and their lifespan, may be stimulated by the volatiles emitted by host plants [68,69]. The perception of volatile molecules by the olive fly’s nervous system can affect the activity of neurotransmitters and various peptides, crucial for homeostatic mechanisms regulating the fly’s lifespan [25,70,71,72].

Our results show that temperature may affect the responses of the olive fly to VOCs. Temperature may have a combined effect on the emission of VOCs from the olives as well as on the tested synthetic chemical VOCs and the flies’ responses. It is known that temperature may indirectly affect herbivorous insects, through interactions with host plants enzymes that synthesize volatile compounds VOCs [73,74]. Also, the temperature shift’s duration influences plant volatiles’ content with short-term changes enhancing the synthesis of volatiles and long-term changes reducing it [75,76]. Yet, temperature affects insects’ fertility, lifespan, and growth rate. Our results show that the olive fly responses to olive fruit volatiles seem to be temperature-dependent. The fly’s responses were more prominent at 30 °C, except for longevity, irrespective of the tested VOCs, while at temperatures above 30 °C, the intensity of the fly’s responses was limited, even in the control ones.

According to our present results, females of the olive fly are active and lay eggs at temperatures, between 17 and 33 °C, but not at 15 and 35 °C. In an earlier work [7], following a different experimental procedure, it was shown that short-term exposure of the fly to high temperatures could adversely affect the longevity and reproduction of the olive fly. In addition, high temperatures can reduce females’ fertility by interfering with oocyte development or causing oocyte degeneration [77]. Wang et al. [51] showed that olive fruit fly fitness was negatively affected after exposure to 35 °C or 37.8 °C by delaying egg maturation and decreasing egg production and lifespan. Yet, Yu et al. [78] showed that temperatures below 10 °C or above 31 °C had a negative effect on the reproduction of *Bactrocera dorsalis* (Diptera: Tephritidae), even causing permanent sterility at extreme temperatures. The effect of temperature on flies’ biology may be explained by the alteration of substances associated with high-temperature resistance (glycose, trehalose, glycogen) [78].

In conclusion, our work shows that a mixture of olive fruit VOCs favored females’ landings on the olive fruits and increased egg production, whereas limonene had a negative effect. This mixture of olive fruit VOCs and limonene seems to have the perspectives of use as, respectively, attractant or repellent products for female olive flies, even in summer, when high temperatures occur in Mediterranean countries. Therefore, our results if corroborated by field experiments may contribute to the improvement of olive fly control using olive fruit volatiles as attractants or repellents for female flies. Further field experiments are currently underway in our laboratory to study the role of the VOCs in the control of the olive fly.

## Figures and Tables

**Figure 1 insects-15-00728-f001:**
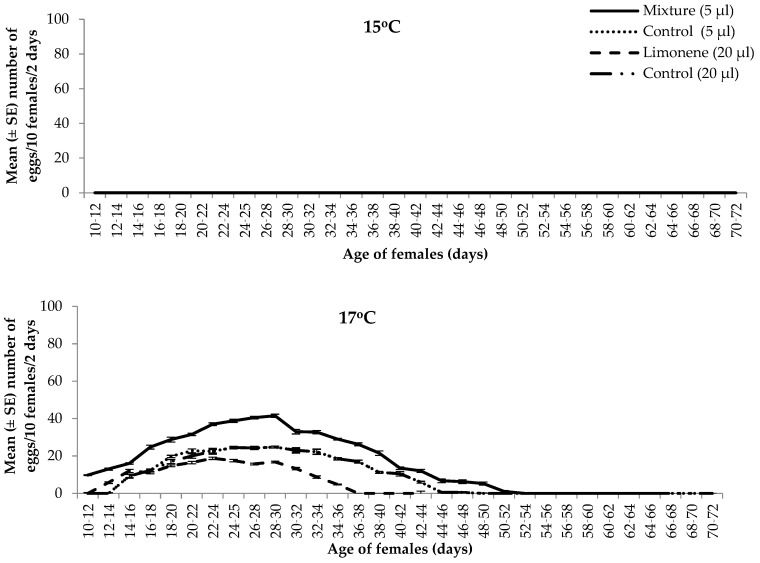
Mean (±SE) number of eggs laid per 10 females of *Bactrocera oleae* per 2 days, in the presence of olive fruit volatile organic compounds (VOCs) at a series of temperatures (15, 17, 20, 25, 30, 33, and 35 °C), 16:8 L:D, 65 ± 1% RH. Mixture 1 of VOCs: *n*-octane–*α*-pinene–limonene–ethyl hexanol–nonanal–*n*-dodecane–decanal–*n*-tetradecane (12:12:2:3:2:1:1:2).

**Table 1 insects-15-00728-t001:** Mean (±SE) number of *Bactrocera oleae* female landings on olives during the photophase period of 16 h (16:8 L:D), in the presence of olive fruit VOCs at a series of temperatures. Ten females were maintained in each cage (replicate) and exposed to certain VOCs or VOC mixtures. Mixture 1: *n*-octane–*α*-pinene–limonene–ethyl hexanol–nonanal–*n*-dodecane–decanal–*n*-tetradecane (12:12:2:3:2:1:1:2), Mixture 2: *n*-octane–ethyl hexanol (4:1), Mixture 3: limonene–nonanal (1:1). There were 4 replicates (cages housing 10 females) for each temperature and VOC. Photoperiod L:D 16:8, relative humidity 65 ± 1%.

Chemicals	Number of Replicates (Ν)	Mean Number of Female Landings on Olive Fruits (±SE)						
		Temperature (°C)						
		15	17	20	25	30	33	35
Mixture 1 (5 μL)	4	0.00 ± 0.00 a^1^A^2^	14.25 ± 0.48 aB	37.25 ± 2.56 aC	53.50 ± 2.21 aD	76.50 ± 1.04 aE	7.25 ± 1.10 aF	0.00 ± 0.00 aA
Mixture 2 (5 μL)	4	0.00 ± 0.00 aA	7.25 ± 0.25 bA	15.00 ± 0.91 bB	28.75 ± 1.03 bC	43.50 ± 2.10 bD	0.00 ± 0.00 bA	0.00 ± 0.00 aA
Mixture 3 (5 μL)	4	0.00 ± 0.00 aA	0.75 ± 0.25 cA	3.50 ± 1.32 cA	7.25 ± 0.62 cB	34.50 ± 1.19 cC	0.00 ± 0.00 bA	0.00 ± 0.00 aA
α-pinene (20 μL)	4	0.00 ± 0.00 aA	2.8 ± 0.85 cA	26.25 ± 1.18 dB	40.50 ± 0.86 dC	59.00 ± 0.40 dD	1.50 ± 0.64 bA	0.00 ± 0.00 aA
limonene (20 μL)	4	0.00 ± 0.00 aA	0.5 ± 0.28 cA	2.00 ± 0.70 cA	2.25 ± 0.25 eA	24.75 ± 1.65 eB	0.00 ± 0.00 bA	0.00 ± 0.00 aA
Control (5 μL)	4	0.00 ± 0.00 aA	1.00 ± 0.00 cA	18.75 ± 2.25 bC	25.75 ± 1.31 bD	54.00 ± 1.08 dE	0.00 ± 0.00 bA	0.00 ± 0.00 aA
Control (20 μL)	4	0.00 ± 0.00 aA	1.00 ± 0.00 cA	18.00 ± 1.50 bB	24.00 ± 1.31 bD	53.00 ± 0.40 dE	0.00 ± 0.00 bA	0.00 ± 0.00 aA

^1^ Means in a column followed by the same lowercase letter are not significantly different (Mann–Whitney *U* test, *p* > 0.05). ^2^ Means in a row followed by the same capital letter are not significantly different (Mann–Whitney *U* test, *p* > 0.05).

**Table 2 insects-15-00728-t002:** Mean (±SE) number of eggs laid per 10 females of *Bactrocera oleae* during their adult life in the presence of olive fruit VOCs at a series of temperatures. Ten females were maintained in each cage (replicate) and exposed to certain VOCs during their adult life. A mixture of VOCs: *n*-octane–*α*-pinene–limonene–ethyl hexanol–nonanal–*n*-dodecane–decanal–*n*-tetradecane (12:12:2:3:2:1:1:2). There were 4 replicates (cages housing 10 females) for each temperature and VOC. Photoperiod 16:8 L:D, relative humidity 65 ± 1%.

Chemicals	Number of Replicates (Ν)	Mean (±SE) Total Number of Eggs/10 Females during Adult Life						
		Temperature (°C)						
		15	17	20	25	30	33	35
Mixture 1 of VOCs (5 μL)	4	0.00 ± 0.00 a^1^A^2^	113.50 ± 3.37 aB	604.00 ± 6.02 aC	632.75 ± 15.22 aD	1348.50 ± 8.99 aE	276.00 ± 7.47 aF	0.00 ± 0.00 aA
Control (5 μL)	4	0.00 ± 0.00 aA	37.75 ± 1.25 bΒ	322.50 ± 5.54 bC	363.00 ± 4.14 bD	999.75 ± 7.28 bE	188.5.00 ± 4.83 bF	0.00 ± 0.00 aA
Limonene (20 μL)	4	0.00 ± 0.00 aA	0.00 ± 0.00 cB	0.00 ± 0.00 cC	78.50 ± 1.04 cD	102.75 ± 8.67 cE	8.75 ± 1.25 cF	0.00 ± 0.00 aA
Control (20 μL)	4	0.00 ± 0.00 aA	35.75 ± 1.43 bB	320.75 ± 4.55 bC	360.75 ± 1.43 bD	994.75 ± 13.00 bE	181.25 ± 2.09 bF	0.00 ± 0.00 aA

^1^ Means in a column followed by the same lowercase letter are not significantly different (Mann–Whitney *U* test, *p* > 0.05). ^2^ Means in a row followed by the same capital letter are not significantly different (Mann–Whitney *U* test, *p* > 0.05).

**Table 3 insects-15-00728-t003:** Longevity (mean ± SE, days) of *Bactrocera oleae* females after exposure to a mixture of olive fruit VOCs or limonene, at a series of temperatures. There were 4 replicates (cages housing 10 females) for each treatment. We scored the number of alive and dead flies every day. A mixture of VOCs: n-octane–α-pinene–limonene–ethyl hexanol–nonanal–n-dodecane–decanal–n-tetradecane (12:12:2:3:2:1:1:2). Photoperiod L:D 16:8, relative humidity 65 ± 1%.

Chemicals	Number of Replicates (Ν)	Mean Longevity (Days ± SE)						
		Temperature (°C)						
		15	17	20	25	30	33	35
Mixture of VOCs (5 μL)	4	68.50 ± 0.50 a^1^A^2^	65.00 ± 0.58 aA	61.50 ± 1.41 aB	55.25 ± 0.48 aC	52.50 ± 0.50 aC	48.50 ± 0.50 aD	11.50 ± 0.96 aE
Control (5 μL)	4	69.00 ± 0.58 aA	65.50 ± 1.26 aB	63.00 ± 0.50 aB	60.00 ± 0.71 bC	56.50 ± 0.96 bD	52.50 ± 0.96 bE	15.00 ± 1.73 aF
Limonene (20 μL)	4	55.50 ± 1.26 bA	44.00 ± 1.41 bB	44.00 ± 0.96 bB	29.25 ± 0.48 cC	27.00 ± 0.58 cD	22.50 ± 0.96 cE	13.00 ± 0.58 bF
Control (20 μL)	4	68.50 ± 0.96 aA	68.50 ± 0.50 aA	63.50 ± 1.00 aA	59.75 ± 0.85 bB	55.50 ± 0.96 bC	52.00 ± 0.82 bC	17.50 ± 0.50 aD

^1^ Means in a column followed by the same lowercase letter are not significantly different (Tukey HSD test, *p* > 0.05). ^2^ Means in a row followed by the same capital letter are not significantly different (Tukey HSD test, *p* > 0.05).

## Data Availability

Data are contained within the article.

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
