# Peer review of "Effect of Olive Fruit Volatiles on Landing, Egg Production, and Longevity of Bactrocera oleae Females under Different Temperatures"

_insects, 2024, doi:10.3390/insects15090728_

Round 1
Reviewer 1 Report
Comments and Suggestions for Authors
The authors investigated the interaction of temperature and volatile organic compounds emitted from olive fruits on longevity, fecundity, and host search behaviour of a serious insect pest, the olive fruit fly, Bactrocera oleae. With this aim, they conducted a series of simple laboratory experiments. The experiments were well designed and conducted. The results of the study can be used for the development of new methods for the monitoring of population density and for the control of this pest. Thus, generally speaking, the manuscript can be published. However, there are some flaws that should be fixed before publication. Two main important problems are:
1) The authors did not explain why they selected namely these volatile organic compounds for their experiments (see my comments to lines 127-139 and 157-159).
2) The authors used Student’s t-test for multiple pairwise comparisons that is not correct (see my comment to lines 177-180).
There are also some minor errors and misprints in the manuscript that should be corrected (see my comments below).
Line 19: Comma is not needed in this sentence.
Line 118: Is “v:w:w 5:4:1” the proportion of weights? If yes, why the first letter is not “w” but“v”? If not, please, explain this abbreviation.
Lines 118-119: This sentence is not quite clear. You mean that the experimental flies were the second generation after the establishing if the laboratory population from wild-collected individuals? If not, how many generations after the establishment were the flies reared in laboratory? Please, indicate this important detail more clearly.
Lines 127-139: In your previous study over 80 different VOCs were detected in the volatile blend emitted from olive fruits (line 304). Please, explain why you used namely these VOCs and these mixtures in these proportions for your experiments.
Line 131: Replace “VOCS” by “VOCs”.
Lines 132-133: You noted that the selection of VOCs for the present study was based on the previous work which showed that these VOCs and their mixtures “favored egg production and mating”. However, your experiments (Table 2) showed that limonene (which is also listed among “the above-mentioned VOCs”) markedly decreased fecundity (see also line 233 and 282-283). Please, explain this controversy.
Lines 157-159: The first experiment showed that 100% limonene had the strongest negative effect on host plant selection. The mixture 3 that included 50% of limonene also had a negative effect, whereas the strongest positive effect was caused by the mixture 1 that, in particular, included about 30% of limonene. Why you did not try to exclude limonene from the mixture 1 to get even stronger positive effect in the second experiment?
Lines 177-180: As clearly indicated in all manuals, Student’s t-test was designed to compare two samples. It is not a correct statistics for multiple pairwise comparisons. For multiple comparisons of the means, one-way ANOVA should be followed by the Tukey HSD test or some other test designed for this purpose. Besides, both t-test and the Tukey HSD test should be used only for normally distributed data. Thus, please, check the distribution of your data and if it is close to normal, use some statistics designed for multiple pairwise comparisons. If the distribution is far from being normal, use some pre-treatment transformation of data or some non-parametric statistics.
Lines 185, 186, 190 etc. “P<0.00” is senseless. P is probability and thus it should be either positive or zero, but not negative. I guess, you mean P<0.01 or P<0.001 or something similar. Please, correct it everywhere in the manuscript.
Lines 202-204: Sorry, but this statement is not true. As seen from Table 1, mixture 3 and limonene significantly decreased the number of landings on the fruits NOT at all of the tested temperatures: at 15, 33 and 35 C (that is about a half of the 7 tested temperatures) the result was zero both for controls and for these VOCs.
Lines 212-214: Student’s t-test should not be used for multiple comparisons in this and other tables (see my comments to lines 177-180).
Line 268: Latin name Bactrocera oleae should be in Italics font.
Line 341: In the study cited as [7] in the list the effect of short-term exposures to extremely high temperatures was studied. This method is quite different from that used in the present study. This should be clearly indicated in the Discussion.
Author Response
|
Comments 1: The authors did not explain why they selected namely these volatile organic compounds for their experiments (see my comments to lines 127-139 and 157-159). |
|
Response 1: In the revised version of our MS, we explained why these volatile organic compounds were selected (page 3; lines 132-137, page 4; lines 166-171).
|
|
Comments 2: The authors used Student’s t-test for multiple pairwise comparisons that is not correct (see my comment to lines 177-180). |
|
Response 2: Following the reviewer’s comments, we revised statistical analysis. The t-test, which was mentioned by error, has been replaced with more appropriate tests designed for multiple comparisons; Tukey HSD for normally distributed data and the non-parametric statistics when the data were far from normally distributed data, even after the pre-treatment transformation (page 4; lines 177-189).
Comments 3: Line 19: Comma is not needed in this sentence. Response 3: We deleted the comma as recommended by the reviewer (page 1; line 19).
Comments 4: Line 118: Is “v:w:w 5:4:1” the proportion of weights? If yes, why the first letter is not “w” but“v”? If not, please, explain this abbreviation. Response 4: We have rewritten this sentence explaining the abbreviation (page 3; line 118).
Comments 5: Lines 118-119: This sentence is not quite clear. You mean that the experimental flies were the second generation after the establishing if the laboratory population from wild-collected individuals? If not, how many generations after the establishment were the flies reared in laboratory? Please, indicate this important detail more clearly. Response 5: In the revised text we clarified that the experimental flies were grown in olives and were descendants of adults emerged in the laboratory from field collected infested olives. (page 3; lines 119-121). Comments 6: Lines 127-139: In your previous study over 80 different VOCs were detected in the volatile blend emitted from olive fruits (line 304). Please, explain why you used namely these VOCs and these mixtures in these proportions for your experiments. Response 6: In the revised MS we explain that we selected to test in our experiments certain VOCs and mixtures, because based on the results of an earlier, they had the most pronounce effects on the flies responses concerning mating percentages and the number of flies landings on the fruit (page 3; lines 132-137, page 4; 166-171). Comments 7: Line 131: Replace “VOCS” by “VOCs”. Response 7: We have replaced “VOCS” by “VOCs” (page 3; lines 131-132). Comments 8: Lines 132-133: You noted that the selection of VOCs for the present study was based on the previous work which showed that these VOCs and their mixtures “favored egg production and mating”. However, your experiments (Table 2) showed that limonene (which is also listed among “the above-mentioned VOCs”) markedly decreased fecundity (see also line 233 and 282-283). Please, explain this controversy. Response 8: We revised this part of our MS clarifying that we tested single VOCs and VOCs mixtures that had the most intense positive or negative effect on egg production and mating, according to the results of our previous work (page 3; lines 128-137). Comments 9: Lines 157-159: The first experiment showed that 100% limonene had the strongest negative effect on host plant selection. The mixture 3 that included 50% of limonene also had a negative effect, whereas the strongest positive effect was caused by the mixture 1 that, in particular, included about 30% of limonene. Why you did not try to exclude limonene from the mixture 1 to get even stronger positive effect in the second experiment? Response 9: The response of the fly to the tested mixtures of VOCs is the result of the combined effects of all the ingredients. A modification of one component does not necessarily result in a corresponding alteration of the overall response observed in the flies. Therefore, we have chosen specific mixtures with distinctive concentrations to test the response of the flies. We could have tested many other different concentrations of VOCs. Specifically, in our experiments we investigated the effects of VOCs based on concentrations identified in our previous research. This unique synthesis of VOCs in mixture 1 was found to have the most positive effects. Modifying the concentration of one VOC may have a positive or negative impact, but this is something to see in the future. Comments 10: Lines 177-180: As clearly indicated in all manuals, Student’s t-test was designed to compare two samples. It is not a correct statistics for multiple pairwise comparisons. For multiple comparisons of the means, one-way ANOVA should be followed by the Tukey HSD test or some other test designed for this purpose. Besides, both t-test and the Tukey HSD test should be used only for normally distributed data. Thus, please, check the distribution of your data and if it is close to normal, use some statistics designed for multiple pairwise comparisons. If the distribution is far from being normal, use some pre-treatment transformation of data or some non-parametric statistics. Response 10: We replaced the t-test, with appropriate tests designed for multiple comparisons. We used Tukey HSD for normally distributed data and the non-parametric statistics when the data were far from normally distributed data, even after the prior transformation (page 4; lines 177-189). Comments 11: Lines 185, 186, 190 etc. “P<0.00” is senseless. P is probability and thus it should be either positive or zero, but not negative. I guess, you mean P<0.01 or P<0.001 or something similar. Please, correct it everywhere in the manuscript. Response 11: We replaced ‘’P<0.00’’ by ‘’P<0.001’’, in all the cases. Comments 12 Lines 202-204: Sorry, but this statement is not true. As seen from Table 1, mixture 3 and limonene significantly decreased the number of landings on the fruits NOT at all of the tested temperatures: at 15, 33 and 35 C (that is about a half of the 7 tested temperatures) the result was zero both for controls and for these VOCs. Response 12: We corrected this statement (page 5; lines 210-213). Comments 13 Lines 212-214: Student’s t-test should not be used for multiple comparisons in this and other tables (see my comments to lines 177-180). Response 13: We corrected it (page 5; lines 222-225, page 6; lines 253-256, page 9; lines 288-290). Comments 14 Line 268: Latin name Bactrocera oleae should be in Italics font. Response 14: We corrected it (page 8; line 281). Comments 15 Line 341: In the study cited as [7] in the list the effect of short-term exposures to extremely high temperatures was studied. This method is quite different from that used in the present study. This should be clearly indicated in the Discussion. Response 15: We rewrote this section, clarifying that the methods of the two studies were different (page 10; lines 351-354). Moreover, we replaced the reference to ‘’Zhao et al.’’ with ‘’ Yu et al.’’ according to the reference list (page 10; line 358).
|
Reviewer 2 Report
Comments and Suggestions for Authors
Dear Dr. Koveos,
your manuscript titled "Effect of fruit volatiles on egg prodution and longevity of Bactrocera oleae females under different temperatures" presents very interesting data that definitely merit attention of a broad readership. The manuscript is well written, experimental methods and statistical analyses are clearly described and can be well understood and followed.
I have only a minor comment/suggestion:
the entire experiment as well as results and conclusions center around the tests and comparison of three different VOC mixtures. I would like to suggest to briefly describe either in the Introduction or in Materials how you came to combine those compounds in the respective mixtures and how the proportions of the particular chemicals in each mixture were defined. I suppose it is the result of one of your previous publications. Please state this fact in more detail in order to be able to follow your reasoning.
Author Response
Comments 1: Dear Dr. Koveos,
your manuscript titled "Effect of fruit volatiles on egg prodution and longevity of Bactrocera oleae females under different temperatures" presents very interesting data that definitely merit attention of a broad readership. The manuscript is well written, experimental methods and statistical analyses are clearly described and can be well understood and followed.
I have only a minor comment/suggestion:
the entire experiment as well as results and conclusions center around the tests and comparison of three different VOC mixtures. I would like to suggest to briefly describe either in the Introduction or in Materials how you came to combine those compounds in the respective mixtures and how the proportions of the particular chemicals in each mixture were defined. I suppose it is the result of one of your previous publications. Please state this fact in more detail in order to be able to follow your reasoning..
Response 1: In our experiments, we studied the effects of certain VOCs and their mixtures based on the concentration found in our previous work (Kokkari et al., 2021). We have explained in the revised text why these volatile organic compounds and mixtures were selected. We have, also, explained why we used these particular proportions for our experiments (page 3; lines 132-137, page 4; 166-171).
Reviewer 3 Report
Comments and Suggestions for Authors
The article addresses an interesting topic in Applied Entomology: the effect of fruit volatiles on landing, egg production and longevity of Bactrocera oleae under different temperatures.
Specific corrections are necessary:
- add "on landing" in the title;
- on lines 23 and 24: add "(depending on the substance tested)" after "decreased";
- line 25: add "except for longevity" after "30 oC";
- lines 35 and 38: repeated section;
- line 42: contact stimuli: explain better;
- Introduction is very long: parts can be moved to Discussion;
- line 121: before being observed for oviposition?
- lines 132-133: better in the Discussion?
- lines 201-204 and 232-234 repeated sections in relation to previous lines;
- line 262: values for 15/17 and 20 oC are statistically different. See capital letters in the Table 3;
- lines 264-265: 13.00 at 35 oC is less than 22.50 at 33 oC;
- line 302: In a previous work: which? [16]?
- lines 336-337: more prominent at 30 oC: (except for longevity)?
- line 344: Yu et al.?
- lines 354-355: if verified by field experiments? If corroborated by? Explain better.
Author Response
|
3. Point-by-point response to Comments and Suggestions for Authors
Reviewer 3 |
|
Comments 1: add "on landing" in the title; |
|
Response 1: We added ‘’on landing’’ in the title of the manuscript (page 1; line 2)
|
|
Comments 2: on lines 23 and 24: add "(depending on the substance tested)" after "decreased"; Response 2: We added ‘’depending on the substance tested’’ after ‘’decreased’’ (page 1; line 25).
Comments 3: line 25: add "except for longevity" after "30 oC"; Response 3: We added ‘’except for longevity’’ after ‘’30 oC’’ (page 1; line 27).
Comments 4: lines 35 and 38: repeated section; Response 4: As suggested by the Reviewer 3, we extirpated lines 35 and 38 and the related references. Accordingly, we modified the number of references in the manuscript and the reference list; now it has been updated to include 79 references instead of 80.
Comments 5: line 42: contact stimuli: explain better; Response 5: We explained it better (pages 1-2; lines 43-44).
Comments 6: Introduction is very long: parts can be moved to Discussion; Response 6: We do not agree with this comment of the Reviewer 3. We believe that our Introduction is not very long and provides a concise and relevant information of the huge amount of the related references. Therefore, we wish to keep our introduction in its present form without any further changes.
Comments 7: line 121: before being observed for oviposition? Response 7: We clarified that olive fruits were maintained at 5±1 °C before they were offered to the flies for oviposition (page 3; line 123-125).
Comments 8: lines 132-133: better in the Discussion? Response 8: We consider that information on the origin of pure chemicals is more appropriately included in the Materials and Methods section of the manuscript (page 3; lines 140-143).
Comments 9: lines 201-204 and 232-234 repeated sections in relation to previous lines; Response 9: These two sections are not repeated results, because in the first one (page 5; lines 211-215 of the revised MS) we describe the effect of VOCs on the number of landings whereas in the second one the respective effect on egg production (page 6; lines 246-248). In addition, the first section has been previously improved based on the comments of the Reviewer 1.
These lines were improved, according to the Reviewer 1 comment. Furthermore, we consider these lines to contribute to a more comprehensive understanding of the results
Comments 10: line 262: values for 15/17 and 20 oC are statistically different. See capital letters in the Table 3; Response 10: We corrected it (page 9; line 277).
Comments 11: lines 264-265: 13.00 at 35 oC is less than 22.50 at 33 oC; Response 11: We corrected it (page 9; lines 279-280).
Comments 12: line 302: In a previous work: which? [16]? Response 12: We added the reference ‘’[15]’’ which is now the number of our previous work (page 10; line 320).
Comments 13: lines 336-337: more prominent at 30 oC: (except for longevity)? Response 13: We added ‘’except for longevity’’ (page 10; line 351).
Comments 14: Yu et al.? Response 14: We replaced ‘’Zhao et al’’ with ‘’Yu et al.’’ (page 10; line 360).
Comments 15: lines 354-355: if verified by field experiments? If corroborated by? Explain better. Response 15: As suggested by the Reviewer 3, we replaced ‘’verified‘’ with ‘’corroborate’’ which explain better the statement (page 11; line 370). |
Round 2
Reviewer 1 Report
Comments and Suggestions for Authors
The authors addressed all comments. The manuscript was substantially improved. I think that it can be published.